# Site-selective protonation enables efficient carbon monoxide electroreduction to acetate

Xinyue Wang[1,2,4], Yuanjun Chen [1,4], Feng Li[3,4], Rui Kai Miao [3,4], Jianan Erick Huang[1], Zilin Zhao[2], Xiao-Yan Li [1], Roham Dorakhan [1], Senlin Chu[2], Jinhong Wu[3], Sixing Zheng[2], Weiyan Ni [1], Dongha Kim[1], Sungjin Park[1], Yongxiang Liang[1], Adnan Ozden[3], Pengfei Ou[1], Yang Hou [2] ✉, David Sinton [3] ✉ & Edward H. Sargent [1] ✉

Electrosynthesis of acetate from CO offers the prospect of a low-carbon-intensity route to this valuable chemical--but only once sufficient selectivity, reaction rate and stability are realized. It is a high priority to achieve the protonation of the relevant intermediates in a controlled fashion, and to achieve this while suppressing the competing hydrogen evolution reaction (HER) and while steering multicarbon ($C_{2+}$) products to a single valuable product--an example of which is acetate. Here we report interface engineering to achieve solid/liquid/gas triple-phase interface regulation, and we find that it leads to site-selective protonation of intermediates and the preferential stabilization of the ketene intermediates: this, we find, leads to improved selectivity and energy efficiency toward acetate. Once we further tune the catalyst composition and also optimize for interfacial water management, we achieve a cadmium-copper catalyst that shows an acetate Faradaic efficiency (FE) of 75% with ultralow HER (<0.2% $H_2$ FE) at 150 mA cm$^{-2}$. We develop a high-pressure membrane electrode assembly system to increase CO coverage by controlling gas reactant distribution and achieve 86% acetate FE simultaneous with an acetate full-cell energy efficiency (EE) of 32%, the highest energy efficiency reported in direct acetate electrosynthesis.

The electrochemical CO reduction reaction (COR) has emerged as a promising approach to achieve the carbonate-free production of multicarbon ($C_{2+}$) products and the decarbonization of chemicals manufacture[1,2]. A long-standing challenge in this area is to suppress the competing hydrogen evolution reaction (HER) and steer selectivity of $C_{2+}$ products to a single valuable product[3,4]. Tuning the stabilities of key intermediates can promote a desired reaction pathway, and strategies have been proposed to suppress HER and improve single-

product selectivity, such as facet regulation[5–7], molecular tuning[8,9], and reactant coverage control[10,11]. Unfortunately, hydrogen evolution (~10%) and other losses limit the efficiency with which CO may be converted to a single product[12–14].

Acetic acid is an industrial solvent and building block for the manufacture of a number of chemicals, polymers, and textiles, and is a precursor to food additives[15,16], with an annual production over 18 million tons and a market size of $13 billion USD/year[17].

[1]Department of Electrical and Computer Engineering, University of Toronto, Toronto, ON M5S 1A4, Canada. [2]Key Laboratory of Biomass Chemical Engineering of Ministry of Education, College of Chemical and Biological Engineering, Zhejiang University, Hangzhou 310027, China. [3]Department of Mechanical and Industrial Engineering, University of Toronto, Toronto, ON M5S 3G8, Canada. [4]These authors contributed equally: Xinyue Wang, Yuanjun Chen, Feng Li, Rui Kai Miao. ✉e-mail: yhou@zju.edu.cn; sinton@mie.utoronto.ca; ted.sargent@utoronto.ca

Today the production of acetic acid relies on fossil fuels, with greenhouse gas (GHG) emissions averaging 1.8 kg CO$_2$-eq/kg acetic acid. Recent analysis suggests that the electrosynthesis of acetate from CO could offer a low-net-emissions approach – but only once sufficient selectivity, reaction rate and stability are realized[18].

Herein we sought a mechanistic understanding of the diversity of C$_2$ products generated through COR. We found that protonation of the relevant intermediates occurs in an uncontrolled fashion on electrocatalysts, leading to the wide distribution of resulting products. We therefore developed a triple-phase interface-engineering strategy to exert control over site-selective protonation of these relevant intermediates to steer COR selectivity toward a single valuable product, seeking to suppress HER completely. We implemented this concept via the regulation of solid/liquid/gas triple-phase interface to achieve atomic-level tuning of the catalyst surface, interfacial water management in the outer Helmholtz plane (OHP), and gas reactant distribution control (Fig. 1). As a result, we report electrosynthesis of acetate with an acetate FE of 86% and acetate full-cell EE of 32%. This energy efficiency surpasses that of the most efficient prior reports in direct acetate electrosynthesis (Supplementary Table 1).

## Results

We began using density functional theory (DFT) to seek a mechanistic explanation for the broad spectrum of C$_2$ products typically witnessed and found that the protonation of relevant intermediates occurs in an uncontrolled fashion (Fig. 2a). We searched the possible reaction pathways for acetate and ethylene formation from CO on a Cu surface, as ethylene is the major competitor to acetate during COR[19,20]. We examined one pathway (path 1) for acetate production and three pathways for ethylene production (paths 2-4) (Fig. 2a). From DFT, path 2 is the most energetically favored route to ethylene on Cu (Supplementary Fig. 1). We noted that the protonation of the ketene intermediate OCCH* is an essential step that determines the formation of acetate vs. ethylene during COR (Supplementary Fig. 2−3). The charge density in the OCCH* intermediate suggests greater charge accumulation with C=O groups and less between Cu and C atoms, indicating stronger C=O and weaker Cu=CH bonds (Fig. 2b). From this we posited that tuning the H affinity to achieve controllable site-selective protonation and protecting the C=O bond intermediates from the H attack could stabilize ketene, an intermediate that steers COR toward acetate production (Fig. 2c).

Single-atom alloy catalysts consist of a dopant metal atomically diluted in the matrix of a host metal, which possesses distinct electronic and geometric features that differ significantly from their constituent metals[21,22]. The well-defined nature of single-atom alloy

catalysts has facilitated theoretical modeling and experimental validation[23,24]. We therefore utilized a strategy based on ideas from single-atom alloying to modulate the H affinity capacities of the catalyst surface by alloying Cu (111) with a metal atom species. We selected a series of metal with different H affinity as the doped atoms[25]. The set of candidate atoms included Pt, Pd, Ni, Co, Zn, Au, Ag, and Cd, candidates for the secondary metal component in Cu bimetallic alloys. Among these, Cd-Cu shows the weakest H adsorption strength, suggesting that it may be a promising candidate (Fig. 2d). Compared with Cu(111), Cd-Cu is found to increase appreciably the barrier of the potential-determining step in COR to acetate vs. ethylene (Fig. 2e and Supplementary Fig. 4). The charge-density-difference calculation reveals that introducing Cd to Cu(111) surface polarizes the electron distribution on those Cu atoms adjacent to the Cd atom (Supplementary Fig. 5). Cd-induced polarization distributes the majority of the electrons between Cd and Cu atoms to form chemical bonds, which reduces the adsorption energy of CHCOH*, an intermediate toward ethylene. Therefore, the weakened H affinity of the Cd-Cu surface to promote site-selective protonation can be attributed to the following aspects. Introducing Cd notably alters the charge distribution among key intermediates, specifically OCCH*, adsorbed on the Cd-Cu alloy surface (Supplementary Fig. 6). The charge density within the OCCH* intermediate indicates a more pronounced charge buildup around C=O groups and a reduced interaction between Cu and C atoms. This reinforces C=O bonds and weakens Cu=CH bonds, facilitating controlled and selective protonation. In addition, Cd exhibits a lower H affinity, as seen in DFT calculations. This facilitates site-selective protonation, preventing C=O bond intermediates from undesired hydrogen attack, and thus stabilizing ketene, a key intermediate that steers COR toward acetate production. Diminished H coverage promotes increased CO coverage on the alloy surface, further suppressing H$_2$ evolution and enhancing acetate production.

Experimentally we synthesized Cd-Cu catalysts, as well as relevant controls. Transmission electron microscopy (TEM) shows tetragonal nanoplates (Fig. 3a and Supplementary Fig. 7), and high-resolution transmission electron microscopy (HRTEM) shows lattice fringes of Cu(111) with lattice parameter 0.21 nm (Fig. 3b). Energy-dispersive X-ray spectroscopy (EDX) elemental mapping indicates that Cd and Cu are evenly distributed to within the 100 nm spatial resolution of the technique (Fig. 3c and Supplementary Fig. 8). Aberration-corrected high-angle annular dark-field scanning transmission electron microscopy (AC HAADF-STEM) shows structure at the atomic scale, resolving Cd (marked by the dashed orange circles) dispersed in Cu (Fig. 3d, e).

We then conducted in situ X-ray absorption spectroscopy (XAS) to examine Cd-Cu further (Supplementary Fig. 9). In situ Cd K-edge

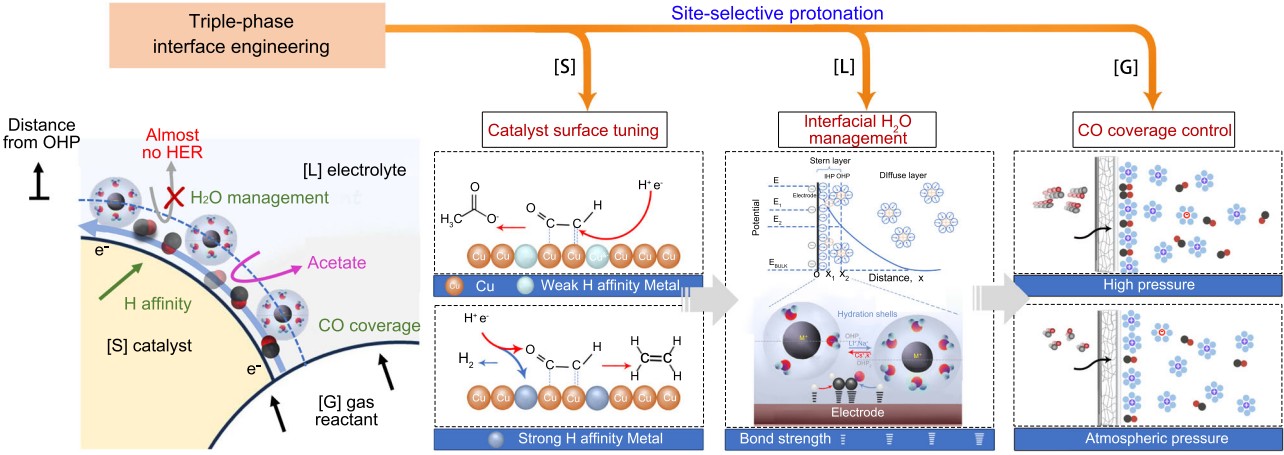

**Fig. 1 | Design principles for site-selective protonation of COR to acetate.** Schematic illustration of solid/liquid/gas triple-phase interface engineering for controllable site-selective protonation for COR to acetate. [S]: solid, [L]: liquid, [G]: gas.

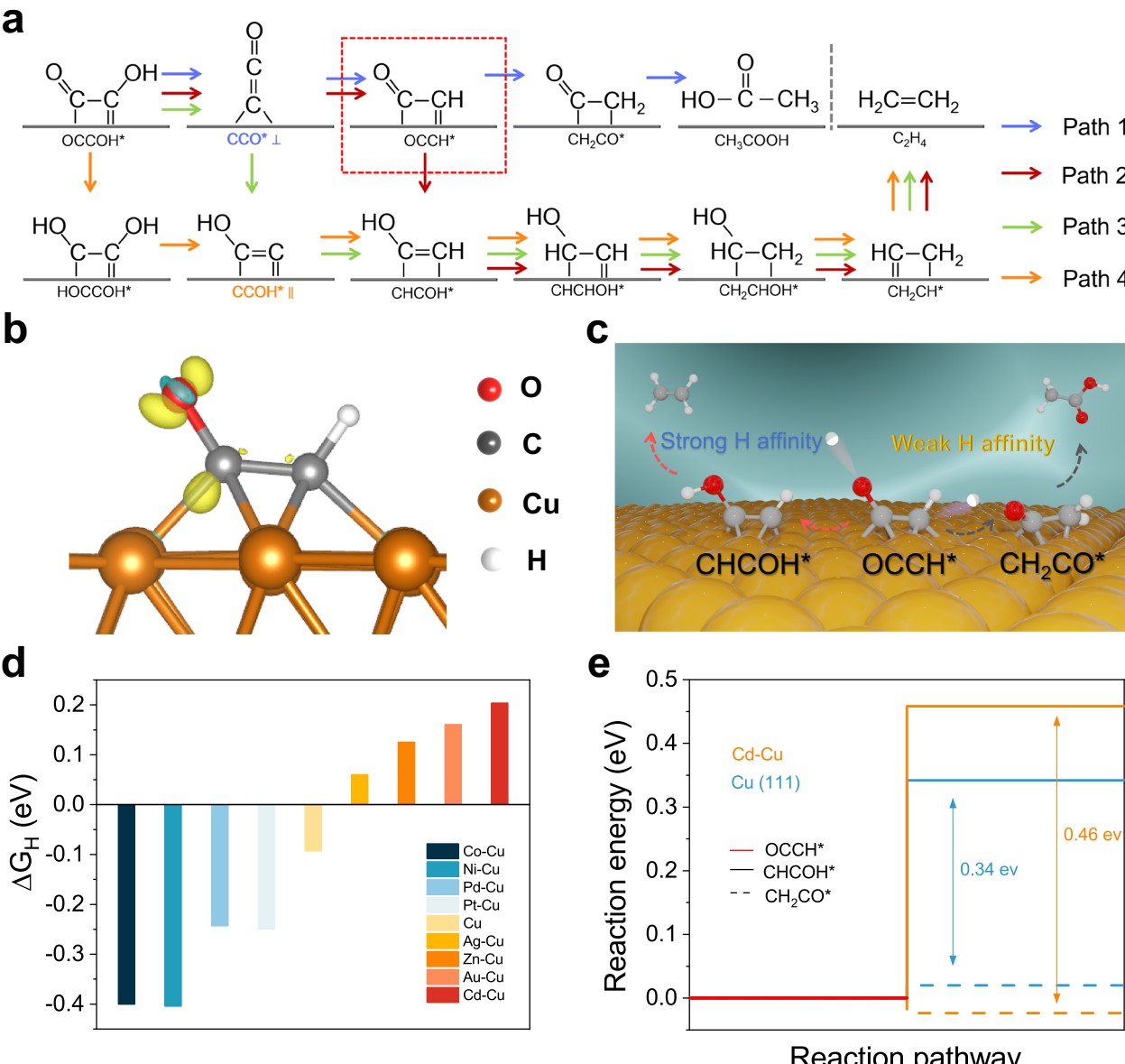

**Fig. 2 | Catalyst design for site-selective protonation of COR to acetate.**
**a** Reaction pathways of COR to acetate and ethylene. **b** Charge density difference of adsorbed OCCH* intermediate on the Cu(111) surface. Yellow and blue contours represent the isosurfaces of electronic charge accumulation and depletion, respectively, with an iso-surface value of 0.03 $e$Å$^{-3}$ implemented. **c** Schematic of key reaction pathways for acetate and ethylene: hydrogenation to CH$_2$CO* for acetate production and hydrogenation to CHCOH* leading to ethylene generation. **d** Gibbs free energy profiles of H adsorption on Cu(111) and X-Cu alloy surfaces (X = Cd, Ag, Au, Zn, Co, Ni, Pd and Pt). **e** Reaction energy diagram for the hydrogenation of Cu=CH bond (OCCH* + H$_2$O + e$^-$ → CHCOH* + OH$^-$) and hydrogenation of C=O bond in OCCH* (OCCH* + H$_2$O + e$^-$ → CH$_2$CO* + OH$^-$) steps on Cd-Cu and Cu(111) models.

extended X-ray adsorption fine structure (EXAFS) spectra show that Cd-Cu exhibits only one dominant peak at 2.28 Å, which is attributed to the Cd-Cu contribution, while the Cd-Cd coordination at 2.79 Å is absent (Fig. 3f, Supplementary Figs. 10, 11 and Supplementary Table 2). Figure 3g shows a maximum wavelet transform of Cd-Cu between 6 and 9 Å$^{-1}$ at a radial distance range of 2–3 Å corresponding to Cd-Cu bonding, which is distinct to that of Cd foil. These results, combined with electron microscopy analysis, suggest an atomic dispersion of Cd in the Cu matrix. The in situ Cu K-edge X-ray absorption near-edge structure (XANES) spectra and EXAFS spectra reveal that the Cu species of Cd-Cu exhibits similar features with Cu foil (Figs. 3h and 3i), suggesting the metallic nature of Cu species as active sites under COR operating condition, consistent with previous reports[26].

We then evaluated the COR performance of the Cd-Cu as well as suitable controls (Co-Cu, Pd-Cu, Au-Cu), then compared these to the

bare Cu control (Supplementary Fig. 12). The Cd loading also was optimized and the optimal loading was determined to be 0.95 wt% with the aid of coupled plasma optical emission spectrometry (ICP-OES) (Supplementary Figs. 13, 14). Cd-Cu exhibited a 97% C$_{2+}$ FE and 86% liquid product FE at 150 mA cm$^{-2}$, higher than the 89% C$_{2+}$ FE and 54% liquid products FE observed on bare Cu (Fig. 4a and Supplementary Fig. 15). The significant increase in the liquid product FE is caused by the shift from ethylene to acetate production. At 150 mA cm$^{-2}$, the ratio of acetate FE (64%) to ethylene FE (10.4%) was 6 on the Cd-Cu, and the acetate FE is 4.8 times higher than that on Cu electrodes (13%) (Fig. 4b and Supplementary Fig. 16). The Cd-Cu suppressed HER to 0.8% H$_2$ FE at 100 mA cm$^{-2}$, while Cu displayed nearly 8 times higher H$_2$ FE at the same current density (Fig. 4c).

To explore the mechanism behind the selectivity change from ethylene to acetate on Cd-Cu, we conducted *operando* attenuated

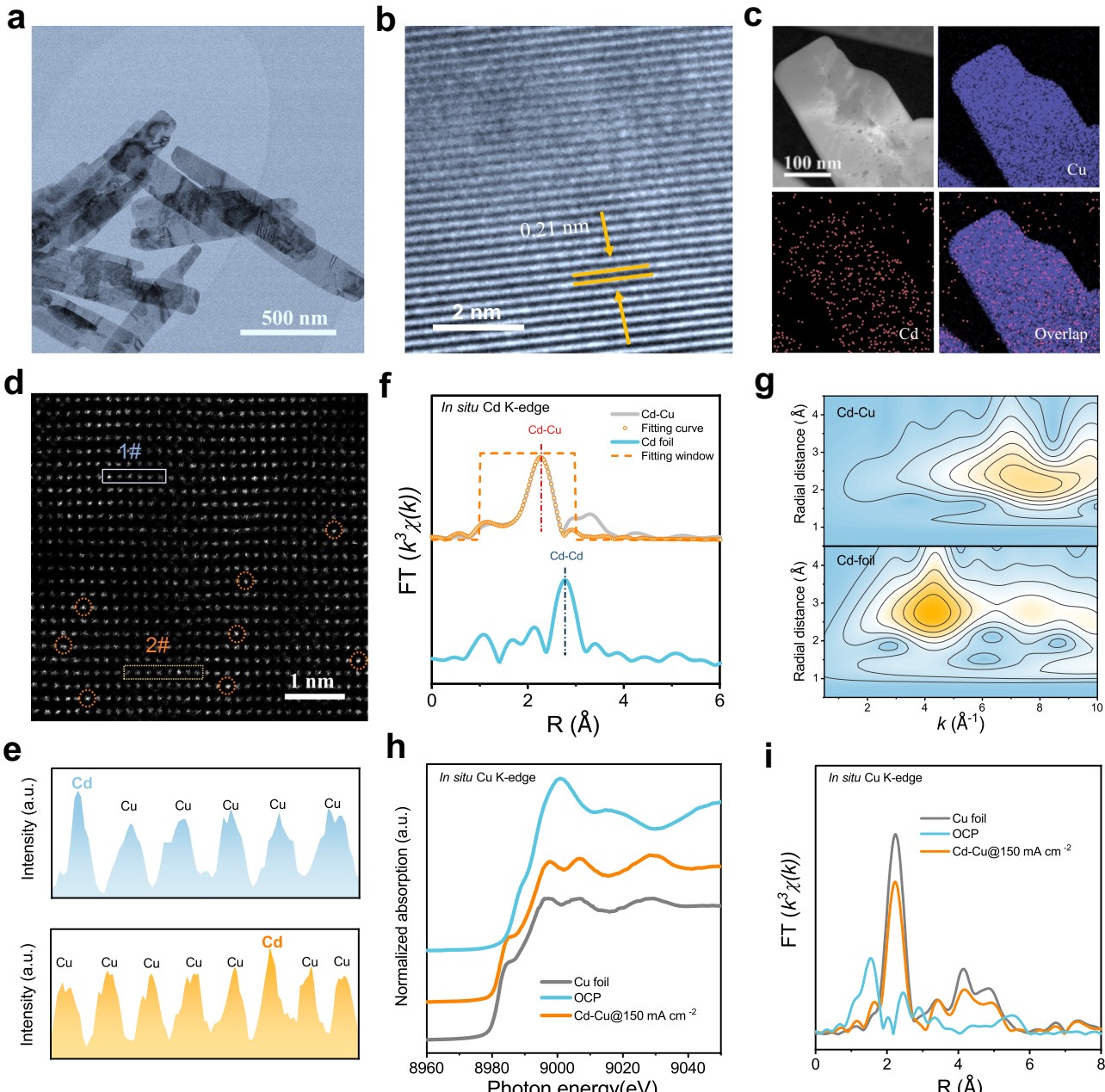

**Fig. 3 | Structural characterization of Cd-Cu catalyst. a** TEM image, **b** HRTEM image and **c** HAADF-STEM image and EDX mapping of Cd-Cu catalyst. **d** AC HAADF-STEM image of Cd-Cu preformed catalyst. Cd atoms are marked using dashed orange circles. **e** The intensity profiles along the light-blue and light-yellow dotted lines in **d**. **f** In-situ Cd K-edge EXAFS spectra. **g** Wavelet transform plots of the Cd-Cu catalyst at 150 mA cm$^{-2}$ (top) and Cd foil (bottom). **h** In-situ Cu K-edge normalized XANES and **i** EXAFS spectra under 150 mA cm$^{-2}$.

total reflection surface-enhanced infrared absorption spectroscopy (ATR–SEIRAS). As shown in Figs. 4d and 4e, we found that the adsorption behavior of *OH (bending vibration of interfacial water molecule) is different for Cd-Cu vs. bare Cu. For Cd-Cu, the peak for *OH was gradually red-shifted from 1637 cm$^{-1}$ at −1.0 V toward the lower wavenumber of 1600 cm$^{-1}$ at −2.0 V. The Stark tuning rate of *OH on Cd-Cu and bare Cu were obtained. A steeper Stark tuning rate indicates an increased sensitivity of the adsorbate to the local electric field of the electrode[27]. In our study, the Stark tuning rate of *OH on the investigated Cu-Cd surfaces falls is ~ 37 cm$^{-1}$/V, a significantly greater value than that observed on bare Cu surfaces, ~ 2 cm$^{-1}$/V. The diminished sensitivity of interfacial water to changes in Cu electrode potential, compared to Cd-Cu electrodes, was ascribed to their indirect connection due to *H

adsorption on the Cu surfaces[28]. In contrast, the increased sensitivity of interfacial water on the Cd-Cu electrode surface was attributed to reduced interference from *H blocking, which is a result of the lower H affinity of Cd. We also investigated the other alloys, and a similar conclusion was reached (Supplementary Fig. 17). These findings are also consistent with Gibbs free energy profiles of HER on Cu(111) and Cd-Cu surfaces (Supplementary Fig. 18), which indicate that the doping Cd onto the Cu(111) surface increases the energy barrier of H$^+$ adsorption step (2H$^+$ –> H* + H$^+$), resulting in reduced H* adsorption on the surface. This shifts the potential-determining step of the HER in the Volmer-Tafel mechanism from the H$_2$ formation and desorption step (2H* –> H$_2$) to the H$^+$ adsorption step. These considerations provide an account for the low HER rate in the Cd-Cu catalyst.

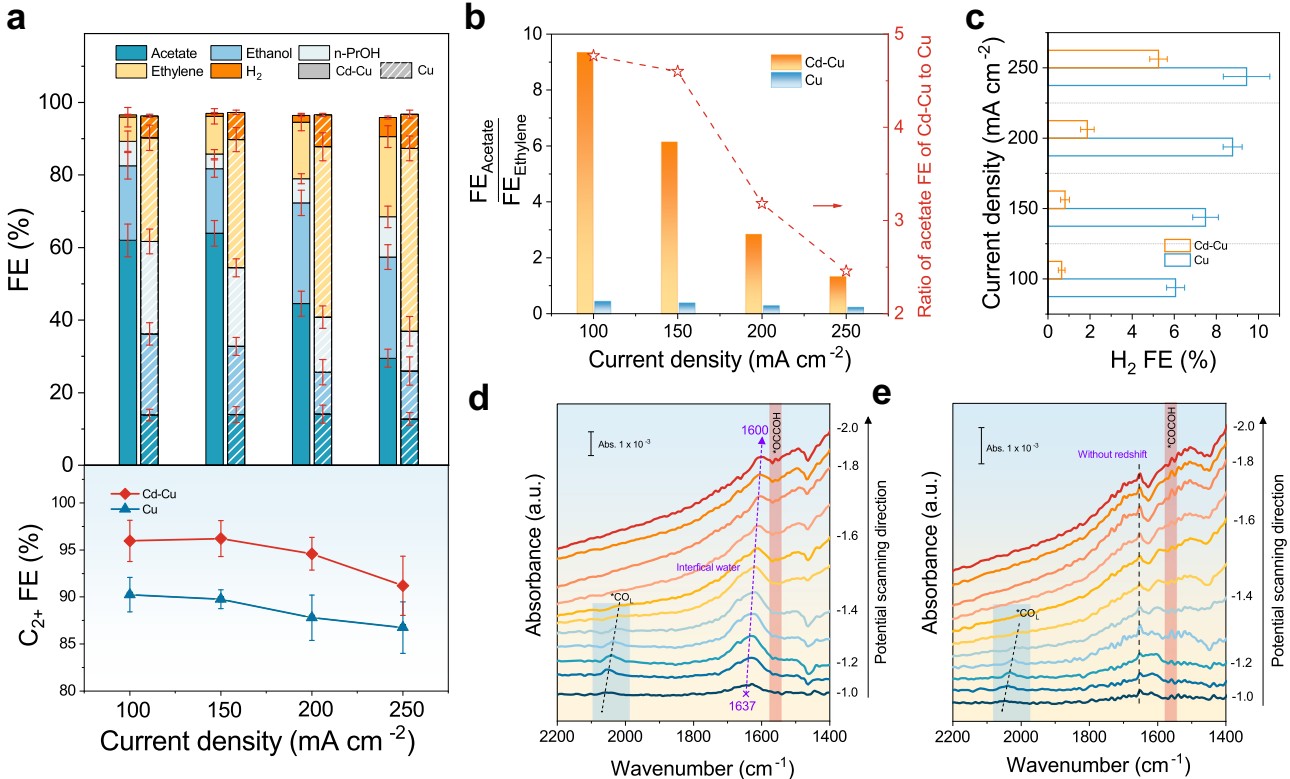

**Fig. 4 | Initial COR performance of the Cd-Cu electrode and *operando* ATR-SEIRAS studies. a** Product distribution in COR (top), $C_{2+}$ products FE (bottom) on the Cd-Cu and the bare Cu under different current density. **b** The ratio of acetate and ethylene FE on Cd-Cu and Cu. **c** $H_2$ FE for Cd-Cu and Cu at different current density in 3 M KOH. The mass loading of the electrodes is 4 mg cm$^{-2}$. **d** *Operando* ATR-SEIRAS spectra of the Cd-Cu electrode. **e** *Operando* ATR-SEIRAS spectra of the Cu electrode. Spectra presented correspond to 64 coadded scans collected with an 8 cm$^{-1}$ resolution. a.u., arbitrary units. Error bars represent the standard deviation of three independent experiments.

This led us to pursue an interfacial water management to further enhance site-selective protonation. We optimized the selection and concentration of alkali metal cations in the electrolyte. Initially, we tested a range of KOH concentrations from 1 M to 5 M. Optimal performance for KOH was in 3 M electrolyte where the peak acetate FE was 64% at 150 mA cm$^{-2}$ (Supplementary Fig. 19). Noting that different hydrated alkali metal cations impact the interfacial water content within the Helmholtz layer, we explored additional cation options (Fig. 5a). In 3 M CsOH electrolyte, the FE towards acetate on Cd-Cu improved to a peak FE of 77% at 50 mA cm$^{-2}$; the FE to acetate decreased significantly and hydrogen evolution increased at current densities over 100 mA cm$^{-2}$. In contrast, the dominant product from COR switched back to ethylene in 3 M NaOH electrolyte, with an acetate FE below 15%. (Fig. 5b and Supplementary Fig. 20). In an effort to achieve higher selectivity towards acetate in the high current density range (>100 mA cm$^{-2}$), we adjusted the ratio of K$^+$ to Cs$^+$ in the electrolyte. As we replaced portions of K$^+$ with Cs$^+$ in the electrolyte (total concentration of 3 M), we observed the suppression of $H_2$ and $C_2H_4$, with their FE dropping to 0.2% and 2.0%, respectively, when we used 2 M K$^+$ and 1 M Cs$^+$ (Fig. 5c and Supplementary Fig. 21). A maximum acetate FE of 75% at partial current density of 113 mA cm$^{-2}$ were observed (Fig. 5d and Supplementary Fig. 22). We also investigated the effect of mixed K$^+$ and Cs$^+$ electrolyte. The addition of Cs$^+$ suppresses hydrogen and ethylene production, leading to enhanced acetate production. The size of hydrated alkali metal cations follows the sequence Cs$^+$(H$_2$O)$_n$ ≤ K$^+$(H$_2$O)$_n$ < Na$^+$(H$_2$O)$_n$, with Cs$^+$ the least hydrated[29,30]. This results in a reduction of interfacial water content within the Helmholtz layer, which diminishes HER activity and reduces C=O protonation in light of the sluggish kinetics of the water activation step. The identity of the alkali metal cation also influences the local electric field, the

sequence Cs$^+$ > K$^+$ > Na$^+$, i.e. with Cs$^+$ inducing the strongest local electric field. (Supplementary Fig. 23 and Supplementary Note 1). Consequently, the intensified local electric field enhances the susceptibility of the Cu−C bond, linked to the electrode surface of the *OCCH intermediate, to hydrogen attack and protonation. This, in turn, contributes to further reducing the tendency of C=O protonation, resulting in the suppression of hydrogen and ethylene production. However, upon complete substitution of K$^+$ by Cs$^+$ (i.e. in pure CsOH electrolyte), the behavior of interfacial water underwent significant changes. *Operando* ATR−SEIRAS indicates three distinct types of interfacial water structures in Cs$^+$-containing electrolytes, including ice-like water, liquid-like water, and free water (Supplementary Fig. 24a). The corresponding area ratios of the three water peaks on Cd-Cu in different electrolytes are depicted in Supplementary Fig. 24b. With the introduction of Cs$^+$, the ratio of free water increased, reaching ~10% in CsOH electrolyte. Therefore, despite Cs$^+$ reducing interfacial water content within the Helmholtz layer, the ratio of free water sharply increases under the influence of the intensified local electric field and applied potential. The presence of free water has been demonstrated to promotes the HER[31,32], offering an account of the observed increase in hydrogen evolution at high current densities.

The Cd-Cu displayed a full-cell potential of -2.04 V at 150 mA cm$^{-2}$, resulting in a full-cell EE to acetate of 29% (Fig. 5e). We further sought to minimize the energy penalty required to separate unreacted CO with the gas products by maximizing the single-pass CO conversion (SPCC) in the system and achieved a SPCC of 90% with a similar product distribution (Fig. 5f). The Cd-Cu electrode provided a new combination of FE, EE, $C_{2+}$ FE and SPCC (Fig. 5g). Through 20 h-electrolysis, the Cd-Cu electrode maintained an acetate FE above 70% (Fig. 5h). Structural characterization of the used Cd-Cu catalyst after a 20 h

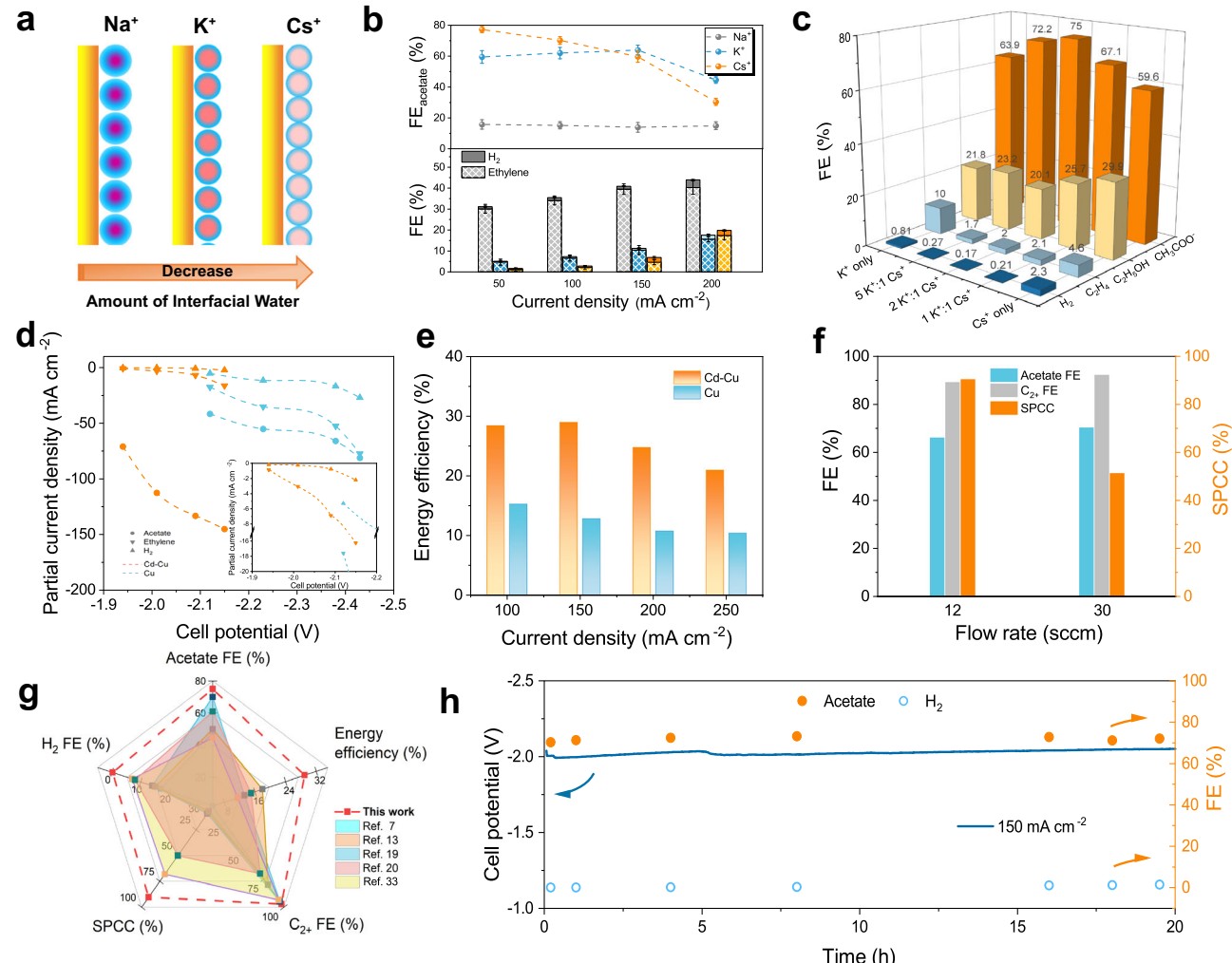

**Fig. 5 | Optimized COR performance of the Cd-Cu electrode. a** Schematic of the amount of interfacial water near the electrode surface with different cation ions. **b** Acetate FE (top) and gas product FE (bottom) on the Cd-Cu in different 3 M electrolytes. **c** FE values for each COR product and $H_2$ in 3 M electrolytes with different ratio of $K^+$ to $Cs^+$. **d** Partial current density of acetate, ethylene and $H_2$. **e** Energy efficiency on the Cd-Cu and the Cu electrodes. **f** Acetate FE, $C_{2+}$ FE and SPCC of the Cd-Cu with different CO flow rate at the current of 2.4 A in an area=16 cm² MEA electrolyser. **g** Comparison of this work with state-of-the-art COR electrodes under atmospheric pressure, including acetate FE, acetate energy efficiency, SPCC, $C_{2+}$ FE and $H_2$ FE. **h** FEs of acetate and $H_2$ as well as cell voltage during 20 h operation of COR at a constant current density of 150 mA cm⁻². The mass loading of the electrodes is 4 mg cm⁻². Error bars represent the standard deviation of three independent experiments. Cell potentials were not iR corrected.

stability test showed that the structure of Cd-Cu was, to within detection and resolution limits, preserved (Supplementary Figs. 25-26).

We noted the importance of regulating reactant coverage on the electrode surface in controlling site-selective protonation, particularly the H coverage[33,34]. This motivated us to adjust the *CO coverage to reduce the concurrent H coverage on the electrode surface. This approach aims to prevent the relevant intermediate from potential H attack. Consequently, a high-pressure membrane electrode assembly (MEA) system was designed (Fig. 6a). When supplying the high-pressure MEA electrolyser with CO gas at 8 bar (Fig. 6b), we achieved a high acetate FE of 86% and full-cell EE of 32%––the highest among previously-published reports (Supplementary Fig. 27 and Supplementary Table 1). To evaluate the economic potential of the acetate electrosynthesis powered by renewable electricity, we carried out a techno-economic analysis (TEA) for the process (Fig. 6c). We took account of the cost of separation, including the protonation of acetate (to acetic acid), liquid and gas product separation (Supplementary Notes 2, 3). The results show that the plant-gate levelized cost for 1

tonne of acetic acid at 150 mA cm⁻² from the present system is projected to be lower than the current average market price of acetic acid.

In this work, we report the design and implementation of a triple-phase interface engineering system to simultaneously achieve tuning of the catalyst surface, interfacial water management, and gas reactant distribution control. This solid/liquid/gas triple-phase interface regulation enables the site-selective protonation of intermediates and favors the stabilization of the ketene intermediate towards acetate. The Cd-Cu single-atom alloy catalyst exhibited an 86% acetate FE and an acetate full-cell EE of 32%, which surpasses the best prior reports. The TEA suggests COR-to-acetate system suggests pathways for technoeconoic positivity for this reaction.

# Methods
## Chemicals
Copper(II) sulfate pentahydrate ($CuSO_4·5H_2O$), Chromium chloride ($CdCl_2$), Sodium hydroxide (NaOH), Potassium hydroxide (KOH), Cesium hydroxide (CsOH), methanol, Nafion perfluorinated resin solution (5 wt% in a mixture of lower aliphatic alcohols and water) were

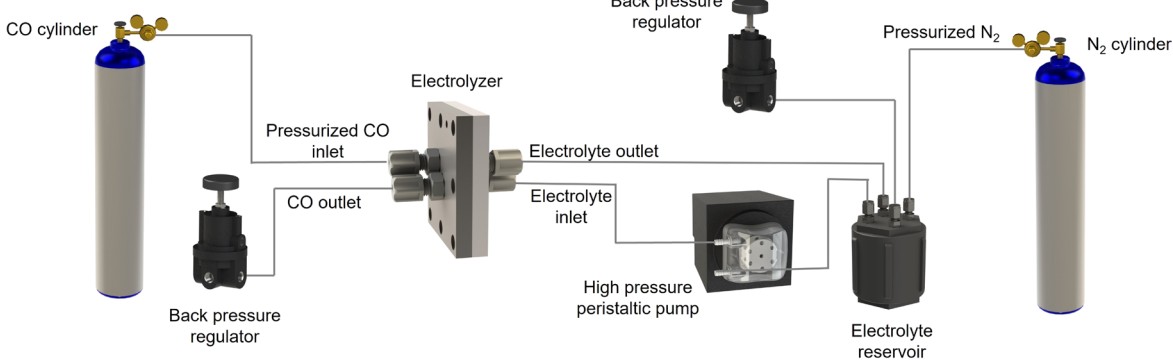

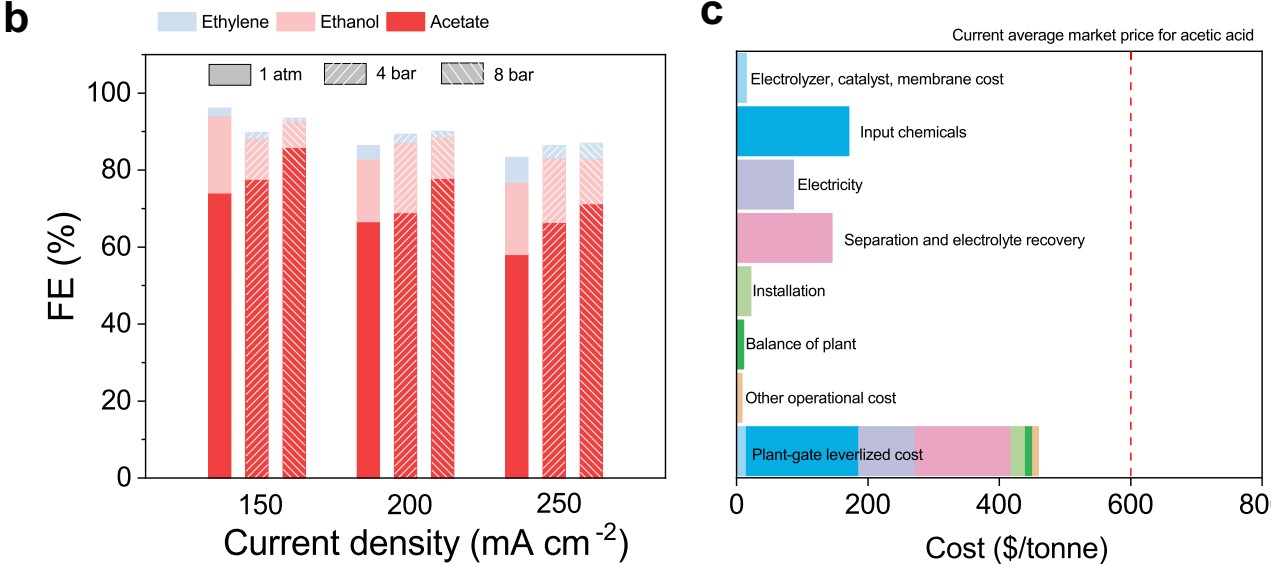

**Fig. 6 | CO electroreduction of the Cd-Cu electrode at high-pressure electrolysis system and technoeconomic analysis. a** Schematic of the high pressure COR system in a membrane electrode assembly electrolyser. **b** Effect of CO pressure on Cd-Cu electrode. The mass loading of the electrodes is 4 mg cm⁻². **c** The TEA is calculated based on the COR performance of Cd-Cu electrode. The total reference price of acetic acid is marked by the red dashed line.

purchased from Sigma Aldrich. Sustainion anion-exchange membrane was purchased from Dioxide Materials. Titanium mesh was received from Fuel Cell Store. The polytetrafluoroethylene (PTFE) gas diffusion layer with 450 nm pore size was obtained from Beijing Zhongxingweiye Instrument Co., Ltd. Copper target (>99.99%) was purchased from Kurt J. Lesker. All chemicals were used without any further purification.

## Synthesis of Cd-Cu catalyst

The Cd-Cu performed material was prepared by hydrothermal method. In a typical procedure, 1.0 g $CuSO_4 \cdot 5H_2O$ and 5.5 mg $CdCl_2$ were dissolved in 50 ml deionized water to form a homogeneous blue solution. The solution was placed in an ice-water bath with vigorous magnetic stirring. 10 ml 1.2 M NaOH solution was dropped into the above solution slowly and stirred half hour continually. Then, keep the mixture refrigerated at 3 °C for 24 hours before transferring it to the Teflon-lined autoclave. The hydrothermal program was set to 130 °C for 18 hours, followed by cooling to room temperature. The Cd-Cu performed material was collected via centrifugation, washed several times with deionized water and dried in vacuum at 60 °C overnight. Finally, the performed material was transferred on GDL to form an electrode. The Cd-Cu catalyst was obtained after electroreduction in 3 M KOH solution at 100 mA cm⁻² for 100 s.

## Electrode preparation

Firstly, Cu coated PTFE membrane (Cu/PTFE) was prepared by sputtering a 200-nm-thick Cu layer onto a PTFE membrane using a 99.99% Cu target (sputtering rate 1 Å/s). Cu/PTFE was used as a conductive gas-diffusion layer for electrode preparation. To prepare the Cd-Cu electrode, 60 mg Cd-Cu catalyst, 4 mL of methanol and 60 μL of 5 wt.% Nafion perfluorinated resin solution were mixed, and then ultrasonicate for 2 h to form a catalyst slurry. By using an airbrush technique and the Cu/PTFE as the gas-diffusion layer, the slurry was sprayed on 3 × 3 cm² Cu/PTFE to achieve Cd-Cu loading of 4 mg cm⁻². The actual loading was determined by weighing the Cu/PTFE before and after spraying. For the Cu electrode, a similar procedure was used to prepare, and the load is also ensured to be 4 mg cm⁻².

## Materials characterization

The morphologies of samples were characterized by field emission scanning electron microscopy (FESEM) (SU8010). Transmission electron microscopy (TEM) and high-resolution transmission electron microscopy (HRTEM) images were collected using JEOL-2100F and Tecnai G2 F20S-TWIN, respectively. The fine morphology was obtained by aberration-corrected scanning transmission electron microscopy (AC-STEM) (JEM-ARM200F working at 300 kV). X-ray powder diffractometer (XRD) measurements were performed in a MiniFlex600 with Cu- Kα radiation. Ex-situ and in-situ X-ray absorption spectroscopy (XAS) data were collected at the 9BM beamline of the Advanced Photon Source (APS, Argonne National Laboratory, Lemont, Illinois). The XAS data were processed using ATHENA and ARTEMIS software incorporated into a standard IFEFFIT package. In situ Raman measurements were conducted in a Renishaw inVia Raman Microscope with a water immersion objective (×63), 785 nm laser. CO was continuously supplied to the gas diffusion electrode during COR.

## Electrochemical measurements

An MEA electrolyser (SKU: 68732; Dioxide Materials) was used, consisting of a gas chamber and an anodic chamber. For membrane electrode assembly electrolyser, the cathode, anion exchange membrane (activated Sustainion membrane), silicone gasket, and IrOx/Ti mesh anode were placed over the gas chamber. In detail, the as-prepared cathode electrode needed to be fixed on the gas chamber by copper tape and make sure it is conducive with the titanium plate. Finally, the electrolyser was assembled with a torque wrench so that the catalyst is stressed uniformly.

The electrochemical measurements were performed with an electrochemical station (Autolab PGSTAT302N) equipped with a 10 A booster (Metrohm Autolab, 10 A). Different ratios of KOH to CsOH solution were used as the anolyte that was fed to the anode chamber at a rate of 10 mL min⁻¹ with a peristaltic pump. CO gas was flowed at 20 sccm by a digital mass flow controller and went through the humidifier, subsequently supplied to the gas chamber. The actual gas flow rate out of the cell under different current was measured with a bubble column.

High-pressure experiments were performed with a setup as shown in Fig. 6a. These employed the same MEA electrolyser and were assembled in the same manner as the experiments at atmospheric pressure. The cathode and anode were separated by an anion exchange membrane (Sustainion). The CO pressure was controlled by a back pressure regulator (SKU: 4783K51, McMaster-Carr) downstream of the cathode gas outlet of the electrolyser. The anolyte (2 M KOH + 1 M CsOH) was pressurized by an inert $N_2$ gas which was then controlled by another pressure regulator at the gas outlet of the electrolyte reservoir. The electrolyte was circulated at the anode by a high-pressure chemical-resistive high-pressure metering pump (SKU: 9154K51, McMaster-Carr). The acetate was assumed to cross over the anion exchange membranes and only the liquid products in the anolyte were sampled upon depressurization of the system. The pressures at the cathode and anode were kept the same throughout the experiments. All pressures reported were absolute pressures ($P_{absolute} = P_{gauge} + P_{atmospheric}$).

In all COR tests, the gas products were analyzed by sampling the outlet stream from the end of the gas chamber with a gas chromatograph (PerkinElmer Clarus 600) coupled with a flame ionization detector (FID) and a thermal conductivity detector (TCD). the liquid products were analyzed by 1H NMR spectroscopy (600 MHz Agilent DD2 NMR Spectrometer) with water suppression mode, using dimethyl sulfoxide (DMSO) as the internal standard and deuterium oxide ($D_2O$) as the lock solvent. Liquid product FE was calculated by collecting products from the anode and cathode sides during the electrolysis.

The SPCC at the conditions of 298.15 K and 101.3 kPa was determined using the following equation:

$$SPCC = (j \times 60\ sec)/(N \times F) \div \left(flow\ rate\left(\frac{L}{min}\right) \times 1(min)/\left(24.05\left(\frac{L}{min}\right)\right.\right. \tag{1}$$

where j is the partial current density of a specific group of products from CO reduction, and N is the electron transfer for every product molecule.

The full-cell EE based on the production of acetate was calculated as follows:

$$EE_{fullcell,Acetate} = \frac{(1.23 + (-E_{Acetate}^0)) \times FE_{Acetate}}{-E_{fullcell}} \tag{2}$$

where $E_{Acetate}^0$ ($E_{Acetate}^0 = 0.454V\ versus\ RHE$) is the thermodynamic potential of CO to acetate, $FE_{Acetate}$ is the measured FE of acetate, and $E_{full\ cell}$ is the full-cell voltage without ohmic loss COR rection evaluated in the MEA electrolyser.

## Computational methods

All the DFT calculations performed in this work were carried out in the Vienna ab initio simulation package (VASP) with a plane wave pseudo-potential implementation[35,36]. The exchange-correlations functional was described by the spin-polarized generalized gradient approximation of Perdew-Burker-Ernzerhof (PBE)[37] and the electron-ion interactions was described by projector augmented wave (PAW) potentials[38]. The kinetic cut-off energy of 450 eV was used for the plane-wave expansion, with Brillouin zone meshed by gamma-point-centered Monkhorst-Pack grids[39]. The long-range Van de Waals interactions were described by the zero-damping DFT-D3 method of Grimme et al.[40]. A charge-density difference was applied to investigate the charge transfer between the adsorbates and the catalysts surfaces.

A hexagonal charged water overlayer-that is, five water molecules and one hydronium ($H_3O^+$)--was introduced to take into consideration of both field and solvation effects[41]. To optimize the structure of the charged water overlayer, Ab initio molecular dynamics simulations were conducted in canonical ensemble (NVT) with the Nose-Hoover thermostat and a 1.0 fs time step at 300 K, as performed in our previous study[42]. Reaction intermediates in COR and HER were included in the optimized geometry from AIMD simulations, and again to perform DFT calculations.

## Data availability

The authors declare that all data supporting the findings of this study are available within the paper and Supplementary Information files. Source data are provided with this paper.

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

## Acknowledgements

The authors acknowledge support from the Ontario Research Fund-Research Excellence program, the National Research Council (NRC), Canada Research Chairs program, and the Natural Sciences and Engineering Research Council (NSERC). Y.H. acknowledges financial support from the National Natural Science Foundation of China (22278364, 22211530045, U22A20432, 22178308). X.W. acknowledges support from the China Scholarship Council (Grant No. 202106320171) and the Zhejiang University Excellent Doctoral Dissertation Funding.

## Author contributions

E.H.S. D.S. and Y.H. supervised the project. X.W. carried out the experiments, analyzed the data and wrote the manuscript. Y.C. conceived the idea, designed the experiments, prepared the catalysts, and wrote the manuscript. F.L. contributed to the DFT calculations. R.K.M. conducted high-pressure system test and technoeconomic analysis. J.E.H. contributed to data analysis and manuscript editing. Z.Z. and S.Z. carried out in-situ ATR-SEIRAS measurements. X.L. and P.O. provided help in DFT calculations. R.D. provided help in NMR analysis. S.C. carried out transmission electron microscopy measurements. J.W. provided help in technoeconomic analysis. W.N., D.K. and S.P. performed XAS measurements. Y.L. and A.O. provided help in membrane-electrode-assembly experiments. All authors discussed the results and assisted during manuscript preparation.

## Competing interests

The authors declare no competing interests.
