## [Peer Review File · Nature Communications]

Site-selective protonation enables efficient carbon monoxide electroreduction to acetateREVIEWER COMMENTS

Reviewer #1 (Remarks to the Author):

In this work, Wang et al. sought a mechanistic understanding of the diversity of C2 products through COR and found an uncontrolled fashion of protonation of the relevant intermediates resulted in wide distribution of products. They developed a triple-phase interface-engineering strategy to achieve controlled site-selective protonation of the relevant intermediates to steer COR selectivity toward acetate and suppress the competing HER. To implement this strategy, they designed a Cd-Cu single-atom alloy catalyst to tune the catalyst surface at the atomic level, regulated water management in the outer Helmholtz plane (OHP) and controlled gas reactant distribution through a high-pressure MEA system design. As a result, they achieved electrosynthesis of acetate with excellent acetate FE of 86% acetate FE simultaneous with a remarkable acetate full-cell energy efficiency of 32%.

Overall, the work is of high novelty and significance and makes a significant advance in COR electrocatalysis. The claims made in this work are well supported by detailed characterization and experiments. The performance achieved in this work is impressive in terms of high FE and energy efficiency as well as ultralow HER (<0.2 % H₂ FE). The catalyst design, fundamental understanding and system engineering in this work will attract extensive attention in material design and electrocatalysis. The manuscript is well organized. The work is strongly recommended for publication subject to the minor revisions below.

1. Cd-Cu single-atom alloy is claimed to display outstanding performance in COR to acetate. What's the Cd loading in the Cd-Cu, has the load of the Cd been optimized?
2. The doping of Cd single atom leads to the obvious suppression of H₂ and promotion of acetate production. The authors should provide more discussion about this result.
3. In terms of the DFT screening for catalyst system, why did the authors select Pt, Pd, Ni, Co, Zn, Au, Ag, Cd as the dopant elements? The authors should provide the premise for such selection and give some explanation.
4. Such low H₂ FE is impressive, and the authors should highlight this point and give further explanation on it.
5. It is suggested that the enhanced selectivity of Cd-doped Cu towards acetate may originate from the modified electronic configuration of Cu resulting from the doping process. To elucidate the connection between electronic structure and acetate selectivity, it is recommended to compare the electronic structure of Cu with and without Cd doping.
6. The high-pressure MEA system shows excellent COR-to-acetate performance. It is recommended to provide a more detailed operating procedure for high-pressure MEA system.

Reviewer #2 (Remarks to the Author):

The authors report on the electroreduction of CO to acetate on an optimized Cu-Cd catalytic system. First an idea regarding speciation in the C₂ pathway is generated via DFT modeling. This is not completely novel, but the way the authors link this to experimental design of the catalyst and process conditions at the catalytic surface is strong. In general, the array of results obtained is impressive and to me it is a bit of a pity that the discussion of the results remain as short as they are. Therefore, for me the main improvements of the manuscript lie in providing a more solid and detailed discussion of the results (for instance, why is a red shift in *OH observed on Cu-Cd?, why does the K⁺/Cs⁺ mixture at this concentration work best and is this dependent on applied current density?) Overall, I think that this will be of interest to the readership of Nature Communications, and I would recommend that the

manuscript can be considered for publication. Below some specific points to improve the present manuscript:

1. First off, the picture quality of figure 1 in the proof was low. I am not sure if this is a result of the conversion of the manuscript upon submission, but would like to ask to double check.
2. The observation that *OH adsorption behavior is different with Cu-Cd than with bare Cu is interesting. Do the authors have any idea why this is the case? Is this a trend that is observed with the other alloys?
3. Although giving interesting results, the rationale behind choosing for a high pressure MEA system compared to the previous system is missing. Could the authors elaborate on this?
4. In general, I am missing information on how stable the single atom doping of Cd is. What is the structure of the catalyst after the reaction? Do you still observe dispersed Cd atoms, or is there any phase segregation?

Reviewer #3 (Remarks to the Author):

1. the Cd-Cu catalyst does not show lattice fringes of Cd monomers, TEM-HR needs to show the lattice fringes of the interface lattice of the two monomers heterostructures of Cd and Cu in order to ensure that the two form an alloy.
2. Stability test in Fig. 5h shows that the target product gradually increases during the first 5 hours, and what is the reason for this?
3. Please check if the iso-surface value symbol in line 199 is formatted correctly.
4. Can the authors compare the acetate FE of Cd-Cu monoatomic alloy catalyst with other reported work in literature.
5. in page 4, line 75-77, it was pointed out that by tuning H affinity to achieve controllable site-selective protonation. The authors should further explain how the 'H affinity' affects the catalytic process. The current hypothesis is not convincing.

Response to Reviewer's Comments

Title: Site-selective protonation enables efficient carbon monoxide electroreduction to acetate

Manuscript ID: NCOMMS-23-43224

We thank the reviewers for their deep engagement with this work and for their advice. We have acted on each comment in this revised work.

A point-by-point response to the comments is provided below. In the revised manuscript, we use blue text to indicate additions and changes made in light of referee advice.

Reviewer #1 (Comments for the Author):

In this work, Wang et al. sought a mechanistic understanding of the diversity of C₂ products through COR and found an uncontrolled fashion of protonation of the relevant intermediates resulted in wide distribution of products. They developed a triple-phase interface-engineering strategy to achieve controlled site-selective protonation of the relevant intermediates to steer COR selectivity toward acetate and suppress the competing HER. To implement this strategy, they designed a Cd-Cu single-atom alloy catalyst to tune the catalyst surface at the atomic level, regulated water management in the outer Helmholtz plane (OHP) and controlled gas reactant distribution through a high-pressure MEA system design. As a result, they achieved electrosynthesis of acetate with excellent acetate FE of 86% acetate FE simultaneous with a remarkable acetate full-cell energy efficiency of 32%.

Overall, the work is of high novelty and significance and make a significant advance in COR electrocatalysis. The claims made in this work are well supported by detailed characterization and experiments. The performance achieved in this work is impressive in terms of high FE and energy efficiency as well as ultralow HER (<0.2 % H₂ FE). The catalyst design, fundamental understanding and system engineering in this work will attract extensive attention in material design and electrocatalysis. The manuscript is well organized. The work is strongly recommended for publication subject to the minor revisions below.

Response: We have acted on each comment and revised the manuscript and Supplementary Information accordingly. Our specific actions taken in response to each comment are detailed below.

Specific comments are as follows:

Comment 1: Cd-Cu single-atom alloy is claimed to display outstanding performance in COR to acetate. What's the Cd loading in the Cd-Cu, has the load of the Cd been optimized?

Response: In the revised work, we determined the Cd loading in the Cd-Cu single-atom alloy using ICP-OES measurement, finding this to be 0.95 wt%. Correspondingly, in this Cd-Cu synthesis process, the feed mass of Cd accounts for 1 wt% of the feed mass of Cu. To optimize Cd loading, we prepared control samples with dosages of 0.5 wt% and 2 wt% respectively. Notably, the 1 wt% sample demonstrated the most favorable acetate production performance. Specifically, on page 6 of manuscript, we now write:

"The Cd loading also was optimized and the optimal loading was determined to be 0.95 wt% with the aid of coupled plasma optical emission spectrometry (ICP-OES) (Supplementary Figs. 13-14)."

Comment 2: The doping of Cd single atom leads to the obvious suppression of H₂ and promotion of acetate production. The authors should provide more discussion about this result.

Response: On revised manuscript page 5, we now more clearly explain the role of Cd doping:

"Introducing Cd notably alters the charge distribution among key intermediates, specifically OCCH*, adsorbed on the Cd-Cu alloy surface (Supplementary Fig. 6). The charge density within the OCCH* intermediate indicates a more pronounced charge buildup around C=O groups and a reduced interaction between Cu and C atoms. This reinforces C=O bonds and weakens Cu=CH bonds, facilitating controlled and selective protonation. In addition, Cd exhibits a lower H affinity, as seen in DFT calculations. This facilitates site-selective protonation, preventing C=O bond intermediates from undesired hydrogen attack, and thus stabilizing ketene, a key intermediate that steers COR toward acetate production. Diminished H coverage promotes increased CO coverage on the alloy surface, further suppressing H₂ evolution and enhancing acetate production."

Comment 3: In terms of the DFT screening for catalyst system, why did the authors select Pt, Pd, Ni, Co, Zn, Au, Ag, Cd as the dopant elements? The authors should provide the premise for such selection and give some explanation.

Response: In the revised work, on page 4, we now more clearly explain the selection of dopant elements. Specifically, we now write:

"We selected a series of metal with different H affinity as the doped atoms.¹ The set of candidate atoms included Pt, Pd, Ni, Co, Zn, Au, Ag, and Cd, candidates for the secondary metal component in Cu bimetallic alloys."

Comment 4: Such low H₂ FE is impressive, the authors should highlight this point and give further explanation on it.

Response: In the revised work, on page 6 and 7, we now more clearly explain the low H₂ FE on Cd-Cu by using *operando* ATR-SEIRAS measurements. We also have highlighted this point. Specifically, we now write:

"The Stark tuning rate of *OH on Cd-Cu and bare Cu were obtained. A steeper Stark tuning rate indicates an increased sensitivity of the adsorbate to the local electric field of the electrode.² In our study, the Stark tuning rate of *OH on the investigated Cu-Cd surfaces falls is $\sim 37 \text{ cm}^{-1}/\text{V}$, a significantly greater value than that observed on bare Cu surfaces, $\sim 2 \text{ cm}^{-1}/\text{V}$. The diminished sensitivity of interfacial water to changes in Cu electrode potential, compared to Cd-Cu electrodes, was ascribed to their indirect connection due to *H adsorption on the Cu surfaces.³ In contrast, the increased sensitivity of interfacial water on the Cd-Cu electrode surface was attributed to reduced interference from *H blocking, which is a result of the lower H affinity of Cd. We also investigated the other alloys, and a similar conclusion was reached (Supplementary Fig. 17). These findings are also consistent with Gibbs free energy profiles of HER on Cu(111) and Cd-Cu surfaces (Supplementary Fig. 18), which indicate that the doping of Cd atom to Cu(111) surface increases the energy barrier of H^+ adsorption step ($2\text{H}^+ \rightarrow \text{H}^* + \text{H}^+$), resulting in reduced H^* adsorption on the surface. This provides an account for the low HER rate in the Cd-Cu catalyst."

Comment 5: It is suggested that the enhanced selectivity of Cd-doped Cu towards acetate may originate from the modified electronic configuration of Cu resulting from the doping process. To elucidate the connection between electronic structure and acetate selectivity, it is recommended to compare the electronic structure of Cu with and without Cd doping.

Response: We have investigated the electronic structure of Cu(111) and Cd-Cu. The DFT calculations indicate that the localization of electron between Cu-Cd atoms induced by Cd doping contribute favourably to acetate production. On revised manuscript page 4, we write:

"The charge-density-difference calculation reveals that introducing Cd to Cu(111) surface significantly polarizes the electron distribution on those Cu atoms adjacent to the doped Cd atom (Supplementary Fig. 5). Cd-induced polarization distributes the majority of the electrons between Cd and Cu atoms to form chemical bonds, which reduces the adsorption energy of CHCOH^* , an intermediate toward ethylene."

Comment 6: The high-pressure MEA system shows excellent COR-to-acetate performance. It is recommended to provide a more detailed operating procedure for high-pressure MEA system.

Response: We have added detailed operating procedure for high pressure MEA system. Specifically, in the *Methods* section, we now write:

"High-pressure experiments were performed with a setup as shown in Fig. 6a. These employed the same MEA electrolyser and were assembled in the same manner as the experiments at atmospheric pressure. The cathode and anode were separated by an anion exchange membrane (Sustainion). The CO pressure was controlled by a back pressure regulator (SKU: 4783K51, McMaster-Carr) downstream of the cathode gas outlet of the electrolyser. The anolyte (2 M KOH + 1 M CsOH) was pressurized by an inert N_2 gas which was then controlled by another pressure regulator at the gas outlet of the electrolyte reservoir. The electrolyte was circulated at the anode by a high-pressure chemical-resistive high pressure metering pump (SKU: 9154K51, McMaster-Carr). The acetate

was assumed to cross over the anion exchange membranes and only the liquid products in the anolyte was sampled upon depressurization of the system. The pressures at the cathode and anode were kept the same throughout the experiments. All pressures reported were absolute pressures ($P_{\text{absolute}} = P_{\text{gauge}} + P_{\text{atmospheric}}$).”

Reviewer #2 (Comments for the Author):

The authors report on the electroreduction of CO to acetate on an optimized Cu-Cd catalytic system. First an idea regarding speciation in the C2 pathway is generated via DFT modeling. This is not completely novel, but the way the authors link this to experimental design of the catalyst and process conditions at the catalytic surface is strong. In general, the array of results obtained is impressive and to me it is a bit of a pity that the discussion of the results remain as short as they are. Therefore, for me the main improvements of the manuscript lie in providing a more solid and detailed discussion of the results (for instance, why is a red shift in *OH observed on Cu-Cd?, why does the K⁺/Cs⁺ mixture at this concentration work best and is this dependent on applied current density?) Overall, I think that this will be of interest to the readership of Nature Communications, and I would recommend that the manuscript can be considered for publication. Below some specific points to improve the present manuscript:

Response: We have revised the work to take account of these advice, as detailed below.

Point 1: why is a red shift in *OH observed on Cu-Cd?

Response: We have added an improved discussion of the red shift in *OH observed on Cu-Cd. The details are shown in the response to specific comment 2.

Point 2: why does the K⁺/Cs⁺ mixture at this concentration work best and is this dependent on applied current density?

Response: We have added an improved discussion of the role of the cation mixture.

In the revised work, we discuss the effect of K⁺/Cs⁺ mixture in light of *operando* ATR–SEIRAS measurements that provide an account of the role of mixed electrolyte. The results show that the choice of the K⁺ and Cs⁺ ratio is influenced by the electric field and interfacial water behavior. On revised manuscript pages 7 and 8, we now write:

"We also investigated the effect of mixed K⁺ and Cs⁺ electrolyte. The addition of Cs⁺ suppresses hydrogen and ethylene production, leading to enhanced acetate production. The size of hydrated alkali metal cations follows the sequence $Cs^+(H_2O)_n \leq K^+(H_2O)_n < Na^+(H_2O)_n$, with Cs⁺ the least hydrated.^{4,5} This results in a reduction of interfacial water content within the Helmholtz layer, which diminishes HER activity and reduces C=O protonation in light of the sluggish kinetics of the water activation step. The identity of the alkali metal cation also influences the local electric field, the sequence $Cs^+ > K^+ > Na^+$, i.e. with Cs⁺ inducing the strongest local electric field. (Supplementary Fig. 23 and Supplementary Note 1). Consequently, the intensified local electric field enhances the susceptibility of the Cu-C bond, directly linked to the electrode surface of the *OCCH intermediate, to hydrogen attack and protonation. This, in turn, contributes to further reducing the tendency of C=O protonation, resulting in the suppression of hydrogen and ethylene production. However, upon complete substitution of K⁺ by Cs⁺ (i.e. in pure CsOH electrolyte), the behavior of interfacial water underwent significant changes. *Operando* ATR–SEIRAS indicates

three distinct types of interfacial water structures in Cs⁺-containing electrolytes, including ice-like water, liquid-like water, and free water (Supplementary Fig. 24a). The corresponding area ratios of the three water peaks on Cd-Cu in different electrolytes are depicted in Supplementary Fig. 24b. With the introduction of Cs⁺, the ratio of free water increased, reaching ~ 10% in CsOH electrolyte. Therefore, despite Cs⁺ reducing interfacial water content within the Helmholtz layer, the ratio of free water sharply increases under the influence of the intensified local electric field and applied potential. The presence of free water has been demonstrated to promote the HER,^{6,7} offering an account of the observed increase in hydrogen evolution at high current densities."

Specific comments are as follows:

Comment 1: First off, the picture quality of figure 1 in the proof was low. I am not sure if this is a result of the conversion of the manuscript upon submission, but would like to ask to double check.

Response: We have corrected the problem of graphical quality throughout.

Comment 2: The observation that *OH adsorption behavior is different with Cu-Cd than with bare Cu is interesting. Do the authors have any idea why this is the case? Is this a trend that is observed with the other alloys?

Response: In the revised work, on pages 6 and 7, we further analyzed the result of *operando* ATR-SEIRAS to clearly explain the ultra low H₂ FE. Specifically, we now write:

The variation in *OH vibrational frequency in response to changes in electrode potential can be attributed to the vibrational Stark effect. "The Stark tuning rate of *OH on Cd-Cu and bare Cu were obtained. A steeper Stark tuning rate indicates an increased sensitivity of the adsorbate to the local electric field of the electrode.² In our study, the Stark tuning rate of *OH on the investigated Cu-Cd surfaces falls is ~ 37 cm⁻¹/V, a significantly greater value than that observed on bare Cu surfaces, ~ 2 cm⁻¹/V. The diminished sensitivity of interfacial water to changes in Cu electrode potential, compared to Cd-Cu electrodes, was ascribed to their indirect connection due to *H adsorption on the Cu surfaces.³ In contrast, the increased sensitivity of interfacial water on the Cd-Cu electrode surface was attributed to reduced interference from *H blocking, which is a result of the lower H affinity of Cd. We also investigated the other alloys, and a similar conclusion was reached (Supplementary Fig. 17). These findings are also consistent with Gibbs free energy profiles of HER on Cu(111) and Cd-Cu surfaces (Supplementary Fig. 18), which indicate that the doping of Cd atom to Cu(111) surface increases the energy barrier of H⁺ adsorption step (2H⁺ → H* + H⁺), resulting in reduced H* adsorption on the surface. This provides an account for the low HER rate in the Cd-Cu catalyst."

On page of 18 in the revised Supplementary Information (Supplementary Fig. 17), we also investigated the *OH adsorption behavior on other alloys. Specifically, we now write:

"We selected Pd-Cu and Au-Cu in light of their weaker and stronger H affinities compared with pure Cu ($\text{Cd} < \text{Au} < \text{Cu} < \text{Pd}$). The Stark tuning rates of $^*\text{OH}$ on Pd-Cu and Au-Cu are $1 \text{ cm}^{-1}/\text{V}$ and $18 \text{ cm}^{-1}/\text{V}$. This trend, where the Stark tuning rate increases as the H affinity decreases, correlates with the selectivity of acetate, is consistent with a picture wherein acetate formation is influenced by the tuning of H affinity."

Comment 3: Although giving interesting results, the rationale behind choosing for a high pressure MEA system compared to the previous system is missing. Could the authors elaborate on this?

Response: On pages of 8 and 9 in the revised manuscript, we have elaborated on the rationale for choosing the high-pressure MEA system as follows:

"We noted the importance of regulating reactant coverage on the electrode surface in controlling site-selective protonation, particularly the H coverage.^{8,9} This motivated us to adjust the $^*\text{CO}$ coverage to reduce the concurrent H coverage on the electrode surface. This approach aims to prevent the relevant intermediate from potential H attack. Consequently, a high-pressure membrane electrode assembly (MEA) system was designed (Fig. 6a)."

Comment 4: In general, I am missing information on how stable the single atom doping of Cd is. What is the structure of the catalyst after the reaction? Do you still observe dispersed Cd atoms, or is there any phase segregation?

Response: We now report characterization of post-operando Cd-Cu samples following stability studies. The results do not evidence any measurable Cd agglomeration, the Cd-Cu structure appearing to be well-maintained.

On page 8 of manuscript, we now write:

"Structural characterization of the used Cd-Cu catalyst after a 20 h stability test showed that the structure of Cd-Cu was, to within detection and resolution limits, preserved (Supplementary Figs. 25-26)."

Reviewer #3 (Comments for the Author):

Specific comments are as follows:

Comment 1: the Cd-Cu catalyst does not show lattice fringes of Cd monomers, TEM-HR needs to show the lattice fringes of the interface lattice of the two monomers heterostructures of Cd and Cu in order to ensure that the two form an alloy.

Response: In the HR-TEM analysis of Cd-Cu catalysts, we observed lattice fringes with a spacing 0.21 nm, corresponding to Cu (111) crystal planes. No lattice fringes associated with Cd species nor with CdCu alloy crystal planes were detected. This observation suggests that the Cd species are highly dispersed on the surface of the Cd-Cu catalyst without any significant influence on the Cu lattice. This finding is in line with previous reports on single-atom alloy catalysts (*Nat. Nanotech.* 2021, **16**, 1386-1393; *Nat. Commun.* 2019, **10**, 5812; *ACS Catal.* 2021, **11**, 1886-1896). We also carried out aberration-corrected high-angle annular dark-field scanning transmission electron microscopy (AC HAADF-STEM) to investigate the dispersion of Cd species at the atomic scale, and this shows individual single Cd atoms dispersed in Cu. In situ XAS measurements show that Cd-Cu shows a single dominant peak at 2.28 Å: this we attribute to the Cd-Cu coordination contribution, while the Cd-Cd coordination at 2.79 Å is not observed. This result from in situ XAS is also consistent with the Cd-Cu catalyst as a single-atom alloy.

Comment 2: Stability test in Fig. 5h shows that the target product gradually increases during the first 5 hours, and what is the reason for this?

Response: In the stability test, the acetate FE was 71% in first 1 hour and 73% after 4 hours. The change of acetate FE during first 5 hours was less than 2%, which is within the margin of measurement error.

Comment 3: Please check if the iso-surface value symbol in line 199 is formatted correctly.

Response: On page 10 of the revised manuscript, the iso-surface value symbol format has been corrected.

Comment 4: Can the authors compare the acetate FE of Cd-Cu monoatomic alloy catalyst with other reported work in literature.

Response: In the revised work, we have provided a table to compare the acetate FE and energy efficiency of Cd-Cu catalyst with other reported work in literature in Supplementary Table 2. The present work achieves an acetate full-cell energy efficiency of 32%, the highest energy efficiency reported in acetate electrosynthesis.

Comment 5: in page 4, line75-77, it was pointed out that by tuning H affinity to achieve controllable site-selective protonation. The authors should further explain how the "H affinity" affects the catalytic process. The current hypothesis is not convincing.

Response: In the revised work, on pages 4 and 5, we now further explain how the H affinity affects the catalytic process. Specifically, we now write:

"Therefore, the weakened H affinity of the Cd-Cu(111) surface to promote site-selective protonation can be attributed to the following aspects. Introducing Cd notably alters the charge distribution among key intermediates, specifically OCCH*, adsorbed on the Cd-Cu alloy surface (Supplementary Fig. 6). The charge density within the OCCH* intermediate indicates a more pronounced charge buildup around C=O groups and a reduced interaction between Cu and C atoms. This reinforces C=O bonds and weakens Cu=CH bonds, facilitating controlled and selective protonation. In addition, Cd exhibits a lower H affinity, as seen in DFT calculations. This facilitates site-selective protonation, preventing C=O bond intermediates from undesired hydrogen attack, and thus stabilizing ketene, a key intermediate that steers COR toward acetate production. Diminished H coverage promotes increased CO coverage on the alloy surface, further suppressing H₂ evolution and enhancing acetate production."

In the revised work, on pages 6 and 7, we also have further analyzed the result of *operando* ATR-SEIRAS to clear explain the effect of H affinity. Specifically, we now write:

"The Stark tuning rate of *OH on Cd-Cu and bare Cu were obtained. A steeper Stark tuning rate indicates an increased sensitivity of the adsorbate to the local electric field of the electrode.² In our study, the Stark tuning rate of *OH on the investigated Cu-Cd surfaces falls is $\sim 37 \text{ cm}^{-1}/\text{V}$, a significantly greater value than that observed on bare Cu surfaces, $\sim 2 \text{ cm}^{-1}/\text{V}$. The diminished sensitivity of interfacial water to changes in Cu electrode potential, compared to Cd-Cu electrodes, was ascribed to their indirect connection due to *H adsorption on the Cu surfaces.³ In contrast, the increased sensitivity of interfacial water on the Cd-Cu electrode surface was attributed to reduced interference from *H blocking, which is a result of the lower H affinity of Cd. We also investigated the other alloys, and a similar conclusion was reached (Supplementary Fig. 17). These findings are also consistent with Gibbs free energy profiles of HER on Cu(111) and Cd-Cu surfaces (Supplementary Fig. 18), which indicate that the doping of Cd atom to Cu(111) surface increases the energy barrier of H⁺ adsorption step ($2\text{H}^+ \rightarrow \text{H}^* + \text{H}^+$), resulting in reduced H* adsorption on the surface. This provides an account for the low HER rate in the Cd-Cu catalyst."

Reference

- 1 Jaramillo, T. F.; Jørgensen, K. P.; Bonde, J.; Nielsen, J. H.; Horch, S.; Chorkendorff, I. *Science* **317**, 100-102 (2007).
- 2 Wang, Y.-H. *et al.* Characterizing surface-confined interfacial water at graphene surface by in situ Raman spectroscopy. *Joule*, **7**, 1-11 (2023).
- 3 Chen, X., Wang, XT., Le, JB. *et al.* Revealing the role of interfacial water and key intermediates at ruthenium surfaces in the alkaline hydrogen evolution reaction. *Nat. Commun.* **14**, 5289 (2023).
- 4 E. V. Vinogradov, P. R. Smirnov, V. N. Trostin. Structure of hydrated complexes formed by metal ions of Groups I—III of the Periodic Table in aqueous electrolyte solutions under ambient conditions. *Russian Chem. Bulletin. Int. Ed.* **52**, 1253-1271 (2003).
- 5 Ma, W. *et al.* Electrocatalytic reduction of CO₂ to ethylene and ethanol through hydrogen-assisted C-C coupling over fluorine-modified copper. *Nat. Catal.* **3**, 478-487 (2020).
- 6 Li, C. Y. *et al.* Unconventional interfacial water structure of highly concentrated aqueous electrolytes at negative electrode polarizations. *Nat. Commun.* **13**, 5330 (2022).
- 7 Shen, L. F. *et al.* Interfacial Structure of Water as a New Descriptor of the Hydrogen Evolution Reaction. *Angew. Chem. Int. Ed.* **59**, 22397-22402 (2020).
- 8 Wei, P. *et al.* Coverage-driven selectivity switch from ethylene to acetate in high-rate CO(2)/CO electrolysis. *Nat. Nanotechnol.* **18**, 299-306 (2023).
- 9 Jin, J. *et al.* Constrained C(2) adsorbate orientation enables CO-to-acetate electroreduction. *Nature* **617**, 724-729 (2023).

REVIEWERS' COMMENTS

Reviewer #1 (Remarks to the Author):

The authors have revised the manuscript accordingly, and I recommend its acceptance for publication.

Reviewer #2 (Remarks to the Author):

The authors have addressed my concerns and have improved the manuscript with additional discussion paragraphs. In my opinion, this manuscript can be accepted for publication.

Reviewer #3 (Remarks to the Author):

The author has supplemented experimental data, and well addressed my questions. I have no further questions/concerns, and recommend it for publication.